# Exploratory search during directed navigation in *C. elegans* and *Drosophila* larva

Mason Klein[1†*], Sergei V Krivov[2†*], Anggie J Ferrer[1], Linjiao Luo[3], Aravinthan DT Samuel[4], Martin Karplus[5,6*]

[1]Department of Physics, University of Miami, Coral Gables, United States; [2]Astbury Centre for Structural Molecular Biology, University of Leeds, Leeds, United Kingdom; [3]Key Laboratory of Modern Acoustics, Ministry of Education, Department of Physics, Nanjing University, Nanjing, China; [4]Center for Brain Science, Department of Physics, Harvard University, Cambridge, United States; [5]Department of Chemistry and Chemical Biology, Harvard University, Cambridge, United States; [6]Laboratoire de Chimie Biophysique, ISIS, Université de Strasbourg, Strasbourg, France

**\*For correspondence:**
klein@miami.edu (MK);
s.krivov@leeds.ac.uk (SVK);
marci@tammy.harvard.edu (MK)

[†]These authors contributed equally to this work

**Competing interests:** The authors declare that no competing interests exist.

**Abstract** Many organisms—from bacteria to nematodes to insect larvae—navigate their environments by biasing random movements. In these organisms, navigation in isotropic environments can be characterized as an essentially diffusive and undirected process. In stimulus gradients, movement decisions are biased to drive directed navigation toward favorable environments. How does directed navigation in a gradient modulate random exploration either parallel or orthogonal to the gradient? Here, we introduce methods originally used for analyzing protein folding trajectories to study the trajectories of the nematode *Caenorhabditis elegans* and the *Drosophila* larva in isotropic environments, as well as in thermal and chemical gradients. We find that the statistics of random exploration in any direction are little affected by directed movement along a stimulus gradient. A key constraint on the behavioral strategies of these organisms appears to be the preservation of their capacity to continuously explore their environments in all directions even while moving toward favorable conditions.

DOI: https://doi.org/10.7554/eLife.30503.001

## Introduction

The trajectories of small organisms often involve stochastic transitions between distinct motor states. A classic example is the swimming behavior of *Escherichia coli* (; *Berg, 1993*), which is characterized by an alternating sequence of runs and tumbles. During runs, the bacteria swim in roughly straight lines, while during tumbles, the bacteria move erratically in place, ultimately picking the direction of a new run at random. The trajectories of larger animals like nematodes and insect larvae are qualitatively similar (*Pierce-Shimomura et al., 1999*; *Luo et al., 2010*). *Caenorhabditis elegans* alternate periods of forward movement with either large angle reorientation maneuvers called pirouettes or small angle turns. Crawling *Drosophila* larvae alternate periods of forward movement with turns (*Lahiri et al., 2011*) where they pause forward motion and use the angle of head swings to pick a new forward orientation.

*C. elegans* also modulates its random exploration in isotropic environments over time (*Wakabayashi et al., 2004*; *Chalasani et al., 2007*). When a worm is placed in a new environment, it first executes a local search, where runs are short. Over time, worms transition to a global search with longer runs. It has been suggested that the transition between local and global searching is

discontinuous (*Calhoun et al., 2014*), and that local search and global search represent two distinct behavioral states.

In stimulus gradients, bacteria, nematodes, and insect larvae bias their random walks toward favorable environments by modulating the statistics of transitions between forward-moving runs and reorientation events. For example, all three organisms exhibit longer runs when pointed toward favorable conditions. Worms and larvae further augment the time spent pointed toward favorable directions by increasing the probability of ending reorientation events with a run pointed in a favorable direction or by gradually steering runs toward favorable directions.

The navigational dynamics of worms and larvae have some parallels with the complex dynamics of a polypeptide chain navigating to the native structure of the protein to which it corresponds. Both are examples of stochastic search processes: the protein needs to fold to the correct native structure, while an organism needs to find food and favorable temperatures, for example. Neither search can be purely random, because it would not be effective. For the protein folding case, it would lead to the Levinthal paradox; that is, it would take an essentially infinite amount of time to fold, while in fact it takes on the order of seconds to minutes (*Zwanzig et al., 1992*; *Karplus, 1997*). The stochastic search of a protein is biased toward the native structure by the potential energy, which is encoded in the sequence as the result of evolutionary selection. The stochastic component of the biased search is necessary to avoid being trapped in local minima on the potential energy surface. Trapping in such metastable states has been observed in protein folding trajectories, where an escape is made possible only by the stochastic nature of the dynamics.

Analogous considerations apply to the navigation dynamics of worms and fly larvae. A purely random search would by very inefficient due to the large size of the space accessible in their normal environment. Thus, living organisms use cues to bias their search. An example is a temperature gradient which plays the role of the potential energy. A purely deterministic search would not be effective here either, because there can be traps (local minima) in the accessible space. These minima could have a physical origin or be due to a complex non-monotonic nature of the cues. A stochastic component in the biased search allows the organisms to overcome the trapping problem. The actual details of the navigational dynamics are specified by the neural circuitry that enervates the muscles. This is optimized by evolution, in analogy to the amino acid sequence in proteins.

The correspondence outlined above suggests that it would be of interest to see whether approaches developed for understanding protein folding dynamics can be used to study the navigational dynamics of worms and larvae. The folding dynamics can be quantitatively described as diffusion (random walk) on a free-energy landscape. In particular, the free energy, $F$, defines the equilibrium probability of the system to be found at a particular position $P \sim exp(-F/kT)$, that is, the system prefers regions with low free energy. The free-energy barrier between unfolded and folded states defines the bottleneck of the folding reaction. The diffusion coefficient describes how quickly the system—whether a worm, larvae, or peptide sequence—explores the configuration space. Together, the free-energy barrier and the diffusion coefficient determine the rate of the process. Such a picture provides a simplified and intuitive, while quantitatively accurate, description of the dynamics. We note also that the free-energy landscape framework is generic and has been successfully applied to many different types of complex dynamics, for example, the dynamics of the game of chess (*Krivov, 2011b*) or patient recovery dynamics after kidney transplant (*Krivov et al., 2014*), as well as to protein folding.

Detailed descriptions of worm or larva dynamics (i.e. how run — turn — run — . . . sequences are chained together) are important to show how complex navigational dynamics are realized in a particular case. However, there are many variants of detailed motions, which are likely to result in very similar larger scale navigational dynamics. Thus, it is of interest to understand and accurately characterize the invariants of large-scale dynamics. It is precisely these invariants that are expected to be optimized by evolution. Moreover, a description making use of the free-energy landscape framework can provide an intuitive picture of the complex navigational dynamics as a whole, versus the localized description of dynamics. It could be used, in particular, to locate equilibrium populations, biases, and bottlenecks during the navigation toward the target in complex environments.

The free-energy landscape of a protein can be determined from long equilibrium trajectories (*Krivov and Karplus, 2004*; *Krivov, 2011a*; *Banushkina and Krivov, 2016*). However, the experimental trajectories of the crawling animals treated here are too short to be considered to be at equilibrium in comparison with those in the reversible folding/unfolding of proteins at equilibrium

(**Shaw et al., 2010**). To determine the equilibrium properties, which are required for the construction of the free-energy landscape, we introduce another general approach to random dynamics, the Markov state model (MSM). This exploits the information contained in a large number of short trajectories measured under identical conditions (**Lane et al., 2011**; **Rao and Caflisch, 2004**; **Krivov et al., 2002**). One refers to the dynamics as Markovian if the next crawling step of an animal depends only on its current spatial position; that is, it quickly forgets the history of its motion. A collection of short trajectories can then be used to determine a probability distribution of future positions of the animal starting from a current position. In particular, the steady state probability distribution can be determined in this way. The analysis is based on the construction of the transition probability matrix, as described in Materials and methods, where it is shown that the steady state distribution is, in fact, the equilibrium distribution for the worms. This matrix provides a complete description of the stochastic dynamics and can be used to determine long time scale behavior.

We first observed the power of this approach when investigating *C. elegans* trajectories in a thermal gradient, with worms placed at their cultivation temperature. It had been thought that the worms would equally avoid both lower and higher temperatures. The analysis of a large number of trajectories with the MSM showed that worms do not strictly avoid warmer temperatures, potentially uncovering a different interpretation of isothermal tracking behavior (see Results section). Encouraged by this result, we extend the protein folding approach for combining trajectories to a more general study of *C. elegans* and *Drosophila* larvae. Specifically, we employ the diffusion coefficient, $D(t)$, which represents the rate of change of the mean square displacement as a function of time for the data set. For both species, in the presence of environmental gradients (e.g. thermal or chemical), it is found that $D(t)$ increases linearly with time for short times (ballistic dynamics), while it approaches a constant value at longer times (stochastic dynamics).

In what follows we investigate the behavior of *C. elegans* and *Drosophila* larvae in the presence of different environmental gradients within this framework. Given the recent interest in search strategies in the absence of information (**Polani, 2009**; **Calhoun et al., 2014**), we also study the motion of both species in a uniform environment (i.e. in the absence of applied gradients).

## Results

### Diffusion and search patterns under isotropic conditions

Navigation in *C. elegans*, *Drosophila*, and other organisms has been treated as a biased random walk (**Berg, 1993**; **Pierce-Shimomura et al., 1999**; **Ryu and Samuel, 2002**), where animals repeatedly transition between bouts of relatively straight forward crawling ('runs') and distinct, often large changes in heading ('turns'). To investigate the relationship between trajectories built in this fashion and more general phenomena of diffusion and Markovian processes, we first studied 2D free crawling behavior in both worm and larva systems in isotropic environments with no applied stimulus. These trajectories (*Figure 1A,A'*) do exhibit diffusive behavior, but do not demonstrate active movement in a particular direction, as demonstrated by their very small values for the dimensionless drift velocity (see Materials and methods).

We observed that worm and larvae dynamics at small time scales are close to deterministic—that is, the animals maintain direction and their trajectories are smooth. At longer time scales movement becomes stochastic or diffusive; in other words, the dynamics in configuration space can be approximately described as Markovian. To estimate the time scale of the transition between these regimes, we inspect the time dependence of the diffusion coefficient, $D(t)$, defined in Materials and methods (*Figure 1B,B'*). For deterministic ballistic dynamics in the $x$ direction, $\Delta x \sim v_x t$ and $D_x(t)$ increases linearly with time, while for diffusive dynamics, $D_x(t)$ is constant. *Figure 1B,B'* suggest that the transition from deterministic to diffusive regimes happens at $t \sim 1000$ s.

*C. elegans* crawling under isotropic conditions drastically reduce their turning rate (i.e. make longer runs) throughout an experiment (*Figure 1B*), as also noted in previous work (**Calhoun et al., 2014**); here, it is studied over a substantially longer time of ~1 hr. We define the turning rate of a population as the total number of turns made, divided by the total time all animals put together spend in forward-crawling runs (i.e. the total time where animals could have turned, but did not—see Materials and methods for details). In particular, the turning rate decreases exponentially with a time constant of approximately 800 s (*Figure 1B*). Inspecting the run durations for individual worm

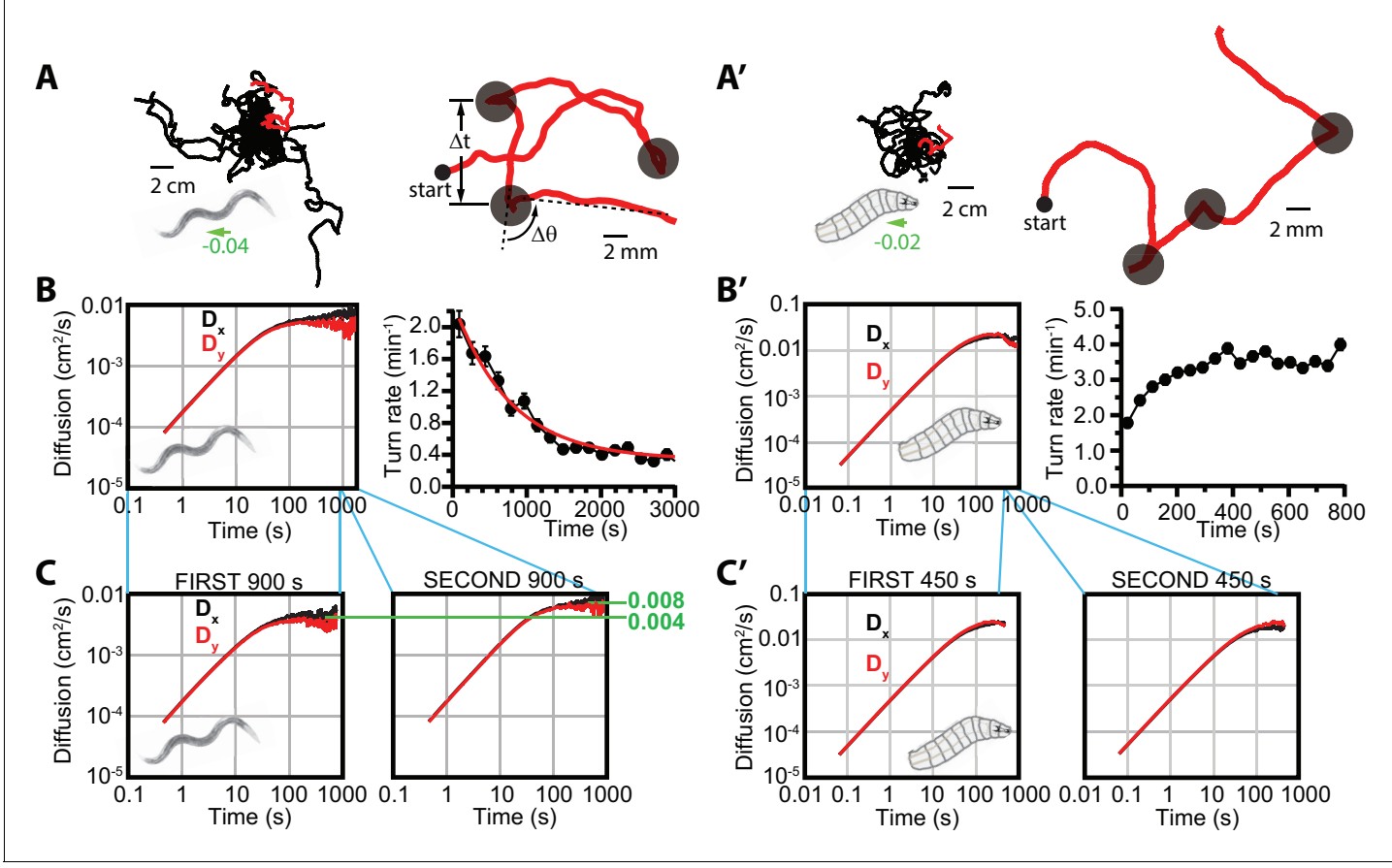

**Figure 1.** Diffusive searching in *C. elegans* and *Drosophila* larva. (**A**) Sample trajectories (left) from 18 worms under isotropic conditions, for 60 min. Tracks have been shifted to all begin at the same location for clarity. A single track (red) is magnified (right) to show the abrupt changes in crawling orientation, flagged as 'turns' (gray circles), with comparatively straight-crawling 'runs' in between. Runs are characterized by their duration $\Delta t$, and turns by their change in heading $\Delta\theta$. (**B**) Diffusion in the *x*- and *y*-directions and turning rate reduction over time. Worms diffuse in both directions (left), while the rate of turning events steadily decreases over 3000 s (right); it fits an exponential decay (red) with a 765 s time constant. (**C**) Diffusion in the *x*- and *y*-directions, splitting the time window into $t < 900$ s (left) and $900 < t < 1800$ s (right). (**A'**) Sample trajectories from 25 *Drosophila* larvae navigating under isotropic conditions for 15 min. (**B'**) Diffusion and turning rate over time for crawling *Drosophila*. $D_x$ and $D_y$ reach similar values, substantially higher than *C. elegans* (left), and the turning rate does not show a dramatic drop, instead increasing for the first ~2 min. before stabilizing. (**C'**) Split $D_x$ and $D_y$ graphs covering the first ($t < 450$ s) and second ($450 < t < 900$ s) halves of the trajectory time, converging to similar values in each. Results for (**B,C**) are based on seven experiments, with 56 tracks and a total of 1608 turns. Results for (**B',C'**) are based on 30 experiments, with 434 tracks and 11,294 turns. The directions of overall population drift for (**A,A'**) are indicated by green arrows, with the numbers indicating the dimensionless drift velocity, in this case extremely small (see Materials and methods). Error bars for (**B,B'**) are ±s.e.m.

DOI: https://doi.org/10.7554/eLife.30503.002

The following source data and figure supplement are available for figure 1:

**Source data 1.** Values and s.e.m. for diffusion coefficient vs. time plots
DOI: https://doi.org/10.7554/eLife.30503.004

**Figure supplement 1.** Consecutive run durations for individual *C. elegans* tracks under isotropic conditions.
DOI: https://doi.org/10.7554/eLife.30503.003

trajectories (***Figure 1—figure supplement 1***), we do not see strong evidence of the turning rate undergoing an abrupt transition from local to global searching, but rather a regular decline, which in the population averages to steady exponential decay.

Turning rate has a clear connection to the dynamics, as frequent turns within a random walk will reduce the diffusion rate. Noting the dramatic reduction in turning rate in ***Figure 1B***, we determined the diffusion coefficients in *x* and *y* from the first part of the experiment, 0 to 900 s, and then separately for the next 900 s. As expected, the diffusion coefficients converge to different limiting values, with diffusion during the second part of the experiments nearly double that during the first part.

This dependence agrees with the simple mean free path estimate of the diffusion coefficient $D \sim v^2/R$, where $v$ is the crawling speed, and the mean turn rate $R$ in the first half of the trajectory is approximately double of that in the second part. We note that $D_x$ and $D_y$ are essentially identical throughout the experiments, only diverging slightly at longer times where the uncertainty has increased (fewer individual tracks last up to 1000 s).

*Drosophila* larvae under isotropic conditions (*Figure 1A'–C'*) exhibit similar behavior in terms of trajectory structure (*Figure 1A'*) and the transition to a diffusive regime, but they do not exhibit a marked decline in turning rate (*Figure 1B'*), which stabilizes after only a few minutes and remains constant throughout their searching behavior. Thus, the transition from local to global searching does not appear on the ~15-min time scales we measured for larva behavior. Given the relatively constant turning rate, it follows that diffusion coefficients calculated for the first (0 to 450 s) and second (450 to 900 s) halves of the experiments converge to similar values. Further, in this larger data set $D_x$ and $D_y$ are essentially identical throughout (*Figure 1C'*). We also note that while $D$ and the turning rate both increase in the first 100 s, the system has not entered a diffusive regime, and the increase in $D$ can be attributed to the ballistic character of the trajectories at this stage.

Taken together, these data show similarity between *C. elegans* and *Drosophila* in the makeup of run — turn — run — ... sequences in crawling behavior, and both conform to a model of diffusion at longer times, but the two animals differ in their long time scale search strategies.

## Diffusion persists alongside thermotaxis and chemotaxis

We next sought to determine what happens to the behavior as animals navigate while exposed to a stimulus along one axis of the crawling surface. Is the diffusive behavior maintained along the axis perpendicular to the stimulus gradient while motion along the parallel axis transforms into a new mode? To investigate this, we observed both *C. elegans* and *Drosophila* navigating along a 1D spatial temperature gradient. The apparatus (*Figure 2A*), as previously described (*Klein et al., 2015*), maintains a stable linear gradient in $x$, and constant temperature in the $y$-direction for fixed $x$-values. Worms cultivated at 15°C and placed in a gradient centered at 20°C exhibited negative thermotaxis (also called 'cryophilic' behavior), while larvae placed at 17.5°C in the same gradient crawled away from cold conditions, exhibiting positive thermotaxis to a preferred range that is independent of cultivation conditions.

*Figure 2B,B'* shows significant diffusion in both $x$ and $y$ directions, even though both types of animals are migrating along the $x$-axis. This suggests that navigation does not eliminate stochasticity along the axis of purposeful navigation. That is, the animals conduct a random search in all directions irrespective of whether they have adopted a target direction. At the same time, the limiting values of $D_x$ and $D_y$ are not equal, with diffusion in the $y$-direction greater than diffusion in the $x$-direction. This suggests that there is a tradeoff between searching along an axis and purposeful travel in that direction.

Both animals move along the $x$-direction toward more favorable environmental conditions by biasing their turning rates. For example, worms undergoing negative thermotaxis reorient their crawling direction more frequently when heading up the temperature gradient, and maintain longer runs when heading down the gradient (*Figure 2B*, lower right). Thermotaxing larvae, similarly, have a higher turning rate when crawling toward aversive colder conditions, and maintain longer runs crawling up the gradient (*Figure 2B'*, lower right). As was true for isotropic conditions, worms decrease their turning rate over time (and larvae maintain a stable level). However, both animals maintain an approximately constant ratio between toward-warm turning rates and toward-cold turning rates. That is, the primary navigational bias that produces thermotaxis is not altered, even at long time scales. Importantly, this supports the method of using early behavior to model long term behavior.

When the dynamics are Markovian, as described in the Introduction, one can use short experimental trajectories to determine the long term equilibrium probability distribution of worms and larvae. *Figure 2C,C'* shows the distribution of both types of animals along the $x$- and $y$-axes, as determined using the Markov state model (see Materials and methods). Distributions are computed using lag times near the transition to diffusive dynamics. As lag time increases, the dynamics become more Markovian and the distributions converge to the limiting distribution (*Figure 2—figure supplement 1B*). The remaining fluctuations are due to relatively small statistics at long times. The limiting

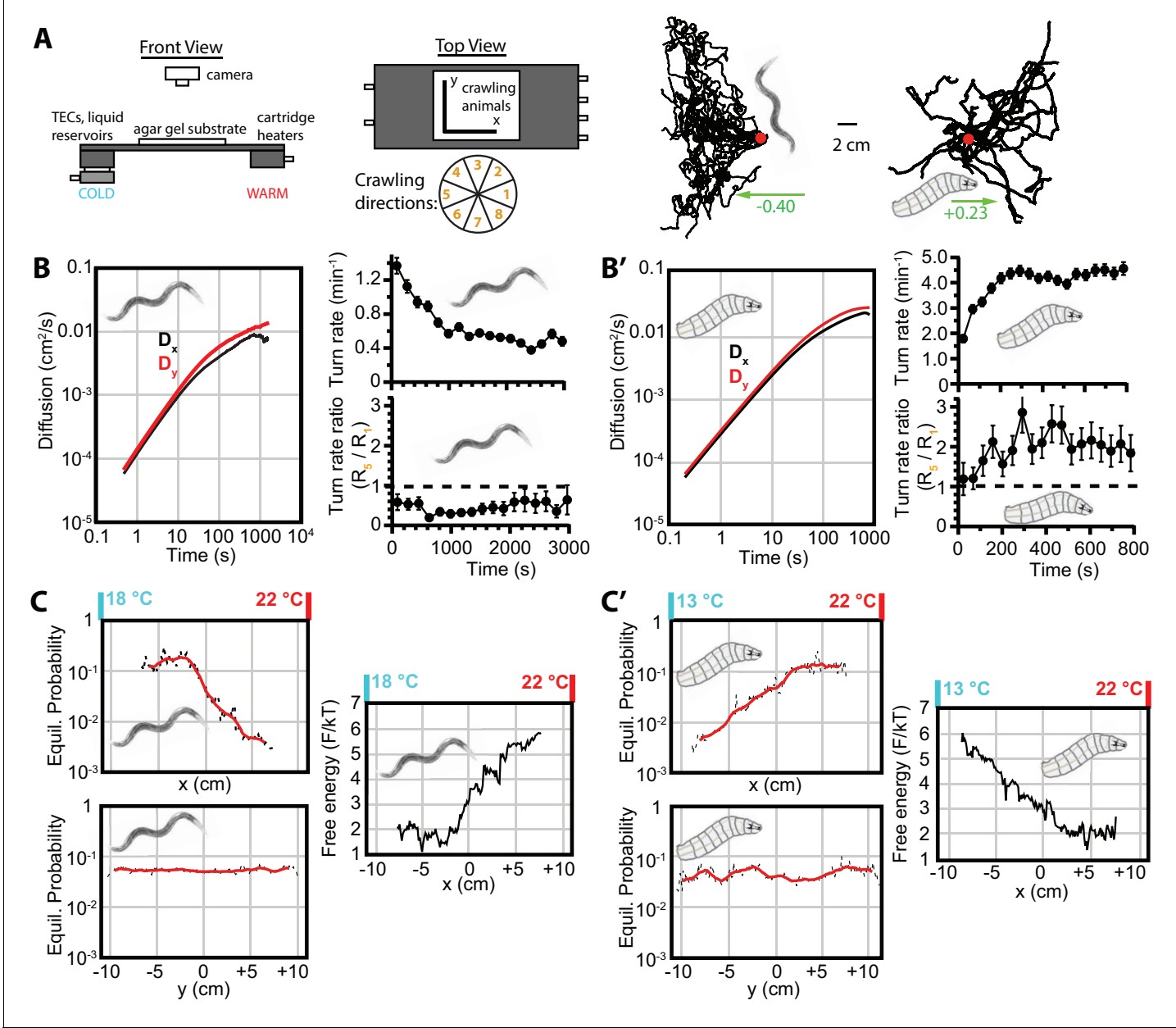

**Figure 2.** Diffusion and navigation in *C. elegans* and *Drosophila* thermotaxis. (**A**) Schematic of the apparatus (left), where animals crawl atop an agar substrate while exposed to a 1D linear temperature gradient. Sample thermotaxis trajectories (right) from 18 *C. elegans* crawling for 60 min and 25 *Drosophila* crawling for 15 min, where the red dots indicate the starting position for all trajectories. Worms start at 20°C and exhibit negative thermotaxis, moving toward their 15°C cultivation temperature, while larvae start at 17.5°C and crawl away from aversive cold temperatures. The wheel indicates labels for crawling direction ranges, with octant 1 parallel to the gradient heading toward warmer temperatures and octant 5 heading antiparallel towards cooler temperatures. (**B**) Diffusion over time (left) in the *x*- and *y*-directions for *C. elegans*. $D_y > D_x$ throughout the experiment, indicating diminished, but highly significant, diffusion along the navigation direction. The average turning rate (upper right) diminishes over time, as in the *Figure 1B,B'* isotropic case; the turning rate ratio $R_5 : R_1$ (lower right) remains below 1 and nearly constant throughout the experiment. (**C**) Equilibrium probability distributions in the *x* (top) and *y* (bottom) directions (both use the same scale), extrapolated from empirical trajectories using the Markov state model (MSM). The lag time used is 750 s. Red lines are smoothed traces to guide the eye. The free-energy picture of equilibrium conditions (right), to place the analysis in context with the protein folding analysis tools employed here. Lower-free energy corresponds to higher population as $P \sim exp(-F/kT)$, where here $kT = 1$. (**B'**) Diffusion, and turning rates for *Drosophila* larvae, also showing $D_y > D_x$. The average turning rate stabilizes early, and the turning rate ratio $R_5 : R_1$, the primary behavioral modulation underlying thermotaxis, remains nearly constant. (**C'**) Equilibrium probability distributions for *Drosophila* larvae (left), and the corresponding free-energy landscape (right), determined from the MSM. Analysis is based on 30 experiments, 131 tracks, and 3061 turns for worms, and 20 experiments, 303 tracks, and 7771 turns for larvae. The directions of overall population drift for (**A**) are indicated by green arrows, with the numbers indicating the dimensionless drift velocity, approximately 10 times greater than for the

*Figure 2 continued on next page*

*Figure 2 continued*

isotropic navigation cases (see Materials and methods). Error bars are ±s.e.m. where shown; in the other cases the error bar size is smaller than the line thickness, and therefore not seen.

DOI: https://doi.org/10.7554/eLife.30503.005

The following source data and figure supplement are available for figure 2:

**Source data 1.** Values and s.e.m. for diffusion coefficient vs. time plots

DOI: https://doi.org/10.7554/eLife.30503.007

**Figure supplement 1.** Reversability, detailed balance, and sampling intervals in the Markov state model approach.

DOI: https://doi.org/10.7554/eLife.30503.006

distribution along the $y$ axis, $P_{eq}(y)$, is constant (up to fluctuations around the boundaries), in agreement with absence of any stimulus along $y$. For worms, the limiting distribution along the $x$ axis, $P_{eq}(x)$, is approximately constant for $T<19.5°$C and then decreases exponentially for $T>19.5°$C. For larvae, the distribution is approximately constant for $T>17.5°$C and then decreases exponentially for $T<17.5°$C. This demonstrates that the worms and larvae diffuse towards $x$ values of their preference, as well as remaining in the regions of their preference, if they are already there. We indicate a more direct connection to the protein folding methods by showing the $x$-distributions in terms of the free energy $F/kT$ (*Figure 2C,C'*, right).

To confirm that these crawling dynamics are not unique to a temperature response, we examined navigation of *C. elegans* exposed to a chemical stimulus corresponding to a 1D linear salt concentration gradient, previously described in *Luo et al. (2014)*. Worms chemotax either up or down salt gradients, depending on the baseline salt level (*Figure 3A*). At low baseline salt concentrations (25 mM), worms move toward higher salt levels, and at high concentrations (75 mM) they crawl down the gradient toward lower salt levels. As with thermal navigation, worm behavior converges to diffusive behavior at longer times (*Figure 3B*), and local searching transitions gradually to global searching via a reduced turning rate (*Figure 3C*). The Markov state model predicts equilibrium population distributions consistent with the net motion of the population (*Figure 3D*).

Despite relatively deterministic movement along one axis, the equilibrium distributions show that the worms and larvae are dispersed over a significant range. This enables them to avoid local 'traps' arising from chemical or thermal cues.

### *C. elegans* diffuse towards warmer temperatures during isothermal tracking

As noted in the Introduction, we applied our diffusion analysis to the distinctive *C. elegans* behavior of isothermal tracking (*Hedgecock and Russell, 1975*; *Luo et al., 2006*). In this behavior, worms placed near their original cultivation temperature ($T_{cult}$) will follow isotherms with extreme precision, indicating a high degree of sensitivity in their thermal response. Since the temperature in our 1D thermal gradient is approximately constant in the $y$-direction, we expect to observe qualitatively different trajectories, with more prominent movement in that direction, and very limited navigation in the perpendicular $x$-direction.

Although we do observe greater diffusion and navigation in the $y$-direction (*Figure 4B*), an examination of $x$-direction navigation revealed a significant asymmetry. The long-term equilibrium position probability distributions ($P_x$ and $P_y$) are approximately constant in $y$, but not in $x$ (*Figure 4C*). In particular, there is an extremely low probability for the worms to be in the $T<T_{cult}$ region, and a substantially higher probability for them to occupy warmer regions. This suggests either a mild preference for $T>T_{cult}$, specific aversion to $T<T_{cult}$, or some other disruption of the traditional interpretation of isothermal tracking behavior.

## Discussion

We studied the navigation of *C. elegans* and *Drosophila* larvae in both isotropic environments and stimulus gradients to assess the relationship between directed movement toward target conditions and the diffusive properties of the overall search patterns. We also studied the negative thermotaxis of *C. elegans* moving toward colder temperatures and the positive thermotaxis of *Drosophila* larvae moving towards warmer temperatures. These behavioral modes represent the better studied forms

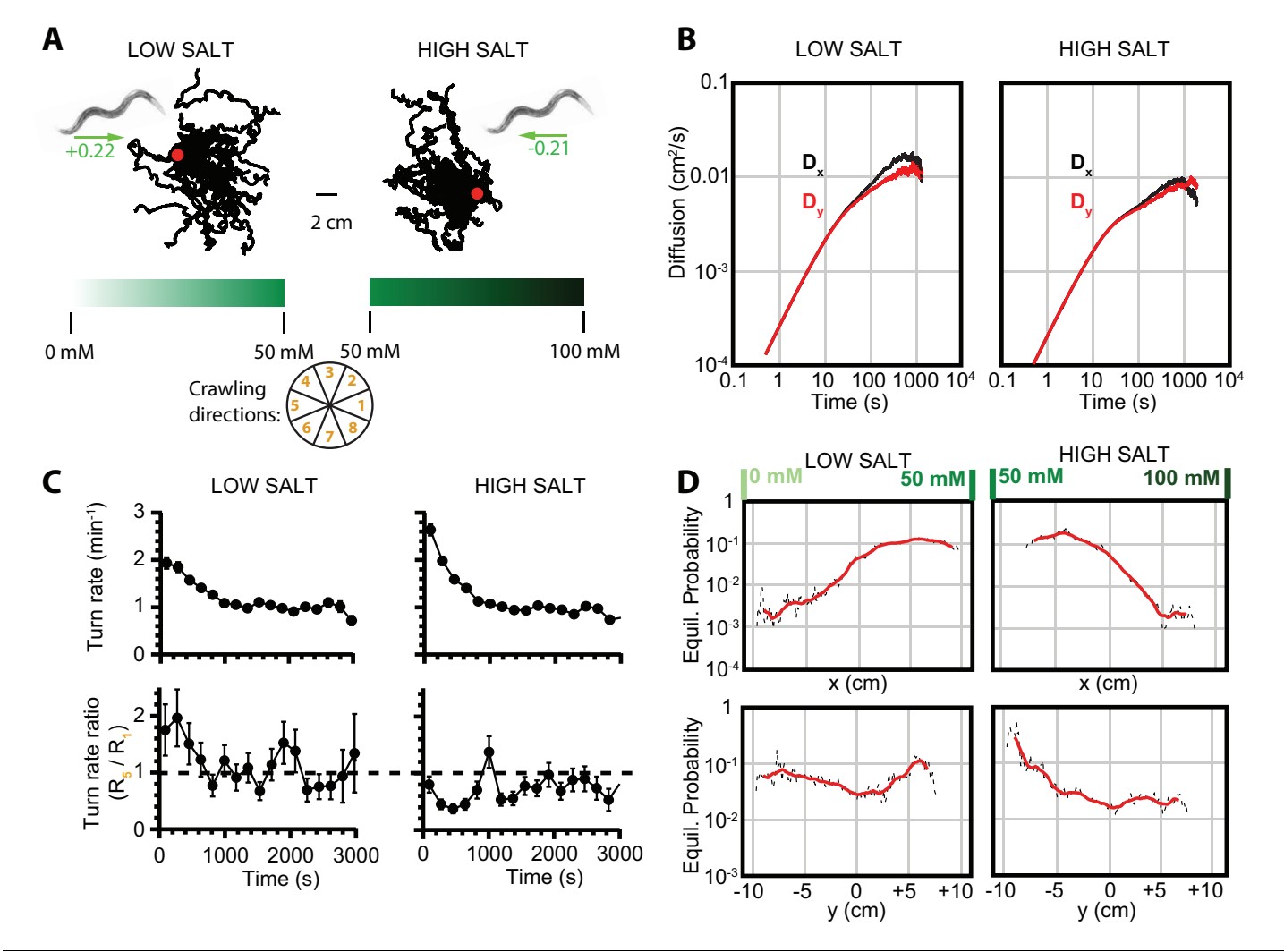

**Figure 3.** Diffusion and navigation in *C. elegans* salt chemotaxis. (**A**) Sample trajectories (25 worms each) of crawling on an agar substrate with a salt concentration gradient increasing toward the right, under a low-salt concentration baseline (25 mM, left) and high-salt concentration baseline (75 mM, right). The wheel indicates labels for crawling direction ranges, with octant 1 pointing directly to higher salt concentrations and octant 5 directly toward low concentrations. The directions of overall population drift are indicated by green arrows, with the numbers indicating the dimensionless drift velocity, approximately 10 times greater than for the isotropic navigation cases (see Materials and methods). (**B**) Diffusion over time in the $x$ and $y$ directions for a low-salt baseline (left) and high-salt baseline (right). Error bars are ±s.e.m. (**C**) Average turning rate across all crawling directions (top) decays over time for both low and high baseline salt concentration gradients. The turning rate ratio between octant 5 (toward lower concentration) and octant 1 (toward higher concentration) does not stabilize as clearly as for thermotaxis, likely indicative of a reduced level of movement toward preferred salt conditions and a smaller data set. Error bars are ±s.e.m. (**D**) Equilibrium probability distributions for worms in low- (left) and high (right)-salt concentration environments, determined by using the Markov state model (MSM). Analysis is based on 14 experiments, 126 tracks, and 4422 turns for low-salt concentration, and 16 experiments, 166 tracks, and 6159 turns for high-salt concentration.

DOI: https://doi.org/10.7554/eLife.30503.008

The following source data is available for figure 3:

**Source data 1.** Values and s.e.m. for diffusion coefficient vs. time plots
DOI: https://doi.org/10.7554/eLife.30503.009

of thermotaxis in these animals. We then examined the ascent and descent of *C. elegans* moving toward preferred salt concentrations.

Treating the motion of small animals in isotropic environments as diffusive random walks is an established method (*Berg and Brown, 1972*; *Berg, 1993*), even yielding analytic solutions under certain conditions (*Lovely and Dahlquist, 1975*). Here, we have focused on diffusion along

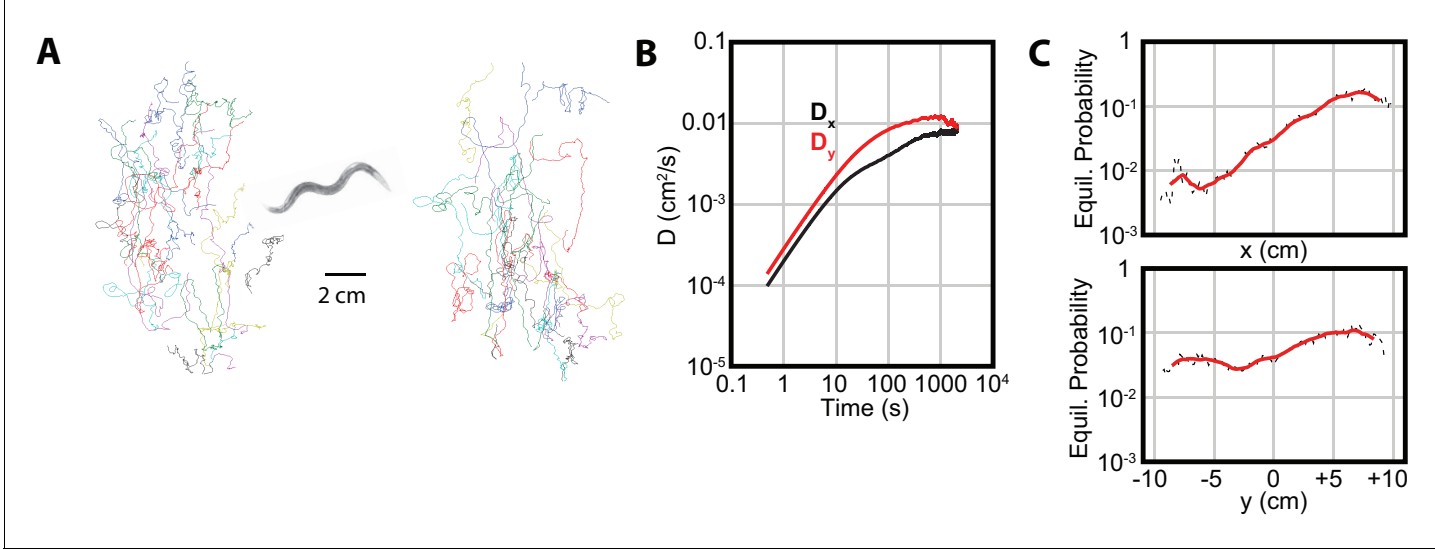

**Figure 4.** Isothermal tracking in *C. elegans* includes diffusion in the *x*-direction. (**A**) Experimental trajectories under temperature gradient conditions, with worms initially placed at their cultivation temperature of 15°C. Colors are used for the reader to distinguish between individual tracks. (**B**) The diffusion constants along the *x* and *y* directions, with $D_y$ dominating, but $D_x$ highly significant. Error bars are $\pm$s.e.m. (**C**) MSM-generated equilibrium probabilities along the temperature gradient (*x* axis) showing the long time scale distribution of the population of worms, and perpendicular to the temperature gradient along isotherms (*y* axis). In *x*, worms avoid regions with low temperatures, but freely explore regions with higher temperature; in *y*, worms explore the axis with approximately equal probability. Analysis based on 25 experiments, with 688 tracks.

DOI: https://doi.org/10.7554/eLife.30503.010

The following source data is available for figure 4:

**Source data 1.** Values and s.e.m. for diffusion coefficient vs. time plots
DOI: https://doi.org/10.7554/eLife.30503.011

perpendicular axes, and used Markov analysis techniques to investigate the combination of exploratory diffusion and targeted navigation. We found that the general framework of diffusion and Markov processes can be used to combine a large number of short trajectories obtained under identical conditions in both isotropic environments and in the presence of stimulus gradients. This approach made it possible for the first time to quantify the statistics of a random search that is concurrent with steady progression towards favorable environments. In both animals and across stimulus types, we found importantly that random exploration in all directions and across all time scales is remarkably robust to progression in a selected direction in a graded environment. That is, the animals undergo diffusive motion (as opposed to ballistic) in both the *x*- and *y*-directions, even during persistent navigation along the *x*-axis. The diffusion coefficients $D_x$ and $D_y$ are not equal during thermotaxis and chemotaxis, but $D$ vs. $t$ plots become approximately constant, indicating a diffusive regime. When nematodes and insect larvae encounter stimuli that bias their random walks in specific directions, the effectiveness of random searching is largely unaffected either parallel or orthogonal to the direction of motion. In these animals, a constraint on the mechanisms that generate navigation in a preferred direction appears to be the preservation of the statistics of random exploration in all directions across time scales. Analysis of the entropy of trajectory configurations, which avoids settling into traps, provides information that is not readily apparent in conventional metrics of drift rates and stimulus-evoked turn rates. Moreover, the approach makes possible large-scale and long time descriptions of the navigational dynamics beyond those available from the standard localized run — turn — run measurements.

We found, in both animal model systems investigated here, that the transition from ballistic to diffusive motion during navigation occurs over a ~1000 s time scale, which is longer than most behavior experiments in studies of these animals. Experiments are typically limited by animals leaving the arena, especially for faster moving late instar *Drosophila* larvae. Combined with the observation, in broad agreement with recent results from other experimenters (*Calhoun et al., 2014*), that the rate of behavioral transitions changes over time (especially in worms), it is possible that further behavioral

transitions at longer time scales have yet to observed. Experimental techniques that enable long-time-scale measurements may be essential for uncovering a more complete picture of the behavior in these animal systems. This would also enable further testing of the probability distributions predicted by the Markov state model.

Additionally, we note that the transition to more global searching (lower turning rate) occurs at very different times in the two model systems under consideration here. We speculate that the much greater mass of the second instar *Drosophila* larva would allow it to delay the transition, as it can afford more time without food. While the global search transition in larvae was not observed on the time scales used here, further experiments could illuminate the issue, such as comparisons in turning rates between fed and starved larvae of the same age—starved animals effectively perform searches, even if not placed in a behavioral arena.

By drawing distinctions between the behavioral transition rates in different crawling directions, we note that the overall changes in the average turning rate (*Figure 2*) are not accompanied by changes in the ratios of the turning rates. This means the navigational bias is preserved, while other aspects of search strategy are modulated. However, the navigational dynamics studied in the cases presented here are rather simple. Consequently, this work may be considered a proof of principle of the utility of employing methods developed for protein folding to understand the behavior of worms and larvae. It will be of interest to study their navigational dynamics in complex conditions with various obstacles, which in the language of protein folding give rise to both enthalpic and entropic barriers. It is important to know where the bottlenecks are in the navigational dynamics towards the target. How the dynamics changes with time (i.e. learning or habituation) in response to different stimuli and different cultivation conditions should also be examined. We expect that to study such questions the description of the dynamics as diffusion on a free-energy landscape will be useful for obtaining a global understanding of the processes involved.

## Materials and methods

### Worm and larva handling

Adult N2 wild-type worms were raised on agar plates (2% wt./vol) with NGM food. For each experiment, around 20 worms (each approximately 1 mm long) were selected under a dissection microscope, rinsed, and placed with a pipette onto the behavior arena in small water droplets. Upon evaporation of the water droplets, the worms began crawling and their movement was recorded.

Wild-type (Canton-S) adult flies were kept in cages (Genesee Scientific) with 6 cm Petri dishes with grape juice and yeast food, with new plates exchanged every 24 hr. Larvae were collected from the plates, with second instar larvae selected by age (24-72 hr AEL) and spiracle development of each individual. The typical larva size at this instar is 1-2 mm in length. For each experiment, between 20 and 30 larvae were rinsed in distilled water, allowed to crawl on agar gel (3% wt./vol) for 5 min, then placed in the behavior arena for video tracking of navigation.

For both worms and larvae, all animals for the experiment are placed on the agar surface together, near the center, with approximately 1 cm separating each animal. Given the small fraction of the available space taken up by the animals, collisions are infrequent. Importantly, when a collision does occur, the event is not flagged as a turn for the purposes of turning rate computation (see below), so if the collision rate decreases over time as animals spread out, the extracted turning rate is not affected.

### Video acquisition and behavioral analysis

A 5 MP CCD camera placed above the arena recorded crawling, with images acquired at 5 Hz. Movies were processed using the MAGAT Analyzer software (*Gershow et al., 2012*), which extracts the position and shape of each animal. Subsequent analysis using custom MATLAB scripts (source code download available, *Source code 1*) segmented the path of each crawling animal into tracks comprised of a sequence of runs (periods of straight crawling) and turns (cessation of forward movement and orientation to a new direction). The run-turn-run-... sequences were used for navigation analysis, and the raw trajectories used for diffusion and Markov state model distributions.

The turning rate describes how often animals alter their crawling direction, and changing turning rate as a function of crawling direction is the primary behavioral modulation that leads to navigation.

We compute the turning rate in the following way. In a given time window, animal $i$ makes $n_i$ turns, with periods of forward crawling ('runs') in between, each run $j$ of duration $\Delta t_j$ (see **Figure 1A,A'**). The total time spent during runs for this animal is $T_i = \sum \Delta t_j$. The turning rate for the individual animal $i$ is $r_i = n_i/T_i$, and the total turning rate for the population during this window is $R = N/T$, where $N = \sum n_i$ and $T = \sum T_i$. In particular, $T$ is the total time where an animal could have turned but failed to do so. In **Figures 2** and **3** turning rates $R_\theta$ are computed for different crawling directions, where only turns and runs that occur within the specified cone of crawling direction are counted.

For a navigation strength metric, we used the dimensionless drift velocity, $<v_x>/<v>$, the average velocity of the population in the $x$-direction, normalized by the overall average speed during runs. This serves as a dimensionless measure of navigation strength. A value of +1 would correspond to every animal crawling directly along the $+x$ direction for the entire experiment; a value of $-1$ would correspond to $-x$ direction crawling; and a value of $0$ would indicate no movement at all, or no bias in crawling direction. For both worms and fly larvae, isotropic conditions result in a very small (order 0.01) navigation strength, while in thermal or chemical gradient environments the navigation strength is of order 0.1. This metric is also employed in (**Luo et al., 2010**; **Gershow et al., 2012**; **Klein et al., 2015**). Green arrows in **Figure 1A,A'**, **Figure 2A**, and **Figure 3A** indicate the navigation strengths for the full population measured.

## Stimulus delivery

For both *Drosophila* larvae and adult *C. elegans*, a temperature-controlled 2D platform established a 1D linear spatial gradient. A large aluminum metal block one each side was maintained at a constant temperature. The cold side was maintained with two thermoelectric coolers (TECs) under PID control, with chilled circulating liquid (a water and anti-freeze mixture) acting as a dissipation reservoir. The hot side was maintained using resistive heaters under PID control. A thin aluminum slab connected the two blocks, which established a smooth linear gradient in the $x$-direction and constant temperature for fixed $x$ in the $y$-direction. An agar gel (3% wt./vol. for larvae, 2% wt./vol. for worms) was placed on the slab. For larva experiments, the temperature across the gel ranged from 13°C to 21°C (17°C in the center, 0.36°C/cm gradient); for worm experiments, the temperature range was 18°C to 22°C (20°C in the center, 0.19°C/cm gradient).

For *C. elegans* experiments using salt concentration gradients, agar gels were poured in two stages to establish a stable, linear salt concentration gradient. We followed the procedure outlined in **Luo et al. (2014)**.

## Determination of steady state and equilibrium probabilities

The equilibrium probabilities ($P_{eq}(x)$ and $P_{eq}(y)$) were computed using the Markov state model (MSM) formalism. To this end, the coordinate (either $x$ or $y$) was partitioned into bins with size $\Delta = 1$ and the numbers of transitions from bin $i$ to bin $j$ after time interval $\Delta t$ ($n_{ji}(\Delta t)$) were computed. The transition probability matrix $P_{ji}(\Delta t)$, the probability to move to bin $j$ from bin $i$ after time interval $\Delta t$ was estimated as $P_{ji}(\Delta t) = n_{ji}(\Delta t)/\sum_j n_{ji}(\Delta t)$. This matrix describes the time evolution of the probability vector as $P_i(t + \Delta t) = \sum_j P_{ij}(\Delta t)P_j(t)$. The stationary, steady state probability distribution $P^{st}$ is computed as the solution of equation $P_i^{st} = \sum_j P_{ij}(\Delta t)P_j^{st}$.

We have also checked whether reversibility and detailed balance are satisfied. First, we computed the steady state fluxes in positive $J^+(x) = \sum_{j<x<i} J_{ij}$ and negative $J^-(x) = \sum_{i<x<j} J_{ij}$ directions, where $J_{ij} = P_{ij}(\Delta t)P_j^{st}$ is the steady state flux from bin $j$ to bin $i$. The fluxes agree with high accuracy (**Figure 2—figure supplement 1A**), meaning that the net flux is zero and we can consider the steady state probability as the equilibrium probability.

The detailed balance, is a more stringent condition, where the fluxes between any two bins must be equal $J_{ij} = J_{ji}$. Due to the limited statistics, and thus higher noise, direct comparison of the fluxes between bins is not informative. We compared a related quantity —the steady state fluxes in positive $J_a^+(x) = \sum_{i,a<x<i} J_{ia} + \sum_{i,i<x<a} J_{ai}$ and negative $J_a^-(x) = \sum_{i,a<x<i} J_{ia} + \sum_{i,i<x<a} J_{ai}$ directions, restricted to transitions to or from a particular node ($a$). The fluxes between bin $x$ and bin $a$ are proportional to the derivatives $d/dx J_a^+(x)$ and $d/dx J_a^-(x)$ and hence from $J_a^+(x) \sim J_a^-(x)$ it follows that $J_{xa} \sim J_{ax}$. **Figure 2** supplemental A compares $J_a^+(x)$ and $J_a^-(x)$ for e.g., $a = 2$. Increasing statistics by considering all the bins in $1.5<a<2.5$ improves the agreement.

The sampling interval (the lag time) $\Delta t$ should be chosen sufficiently large so that the dynamics become Markovian. *Figure 2* supplemental B shows how with increasing lag time the determined equilibrium probabilities converge to the limiting one.

Inclusion of other parameters such as the body angle and whether the animals were stationary or moving did not significantly change the results.

## Determination of diffusion coefficients

For flat free-energy profiles, with no drift term, $F(x) \sim \mathrm{const}$, the diffusion coefficient can be estimated as $D_x(t) = \frac{1}{2}\langle \Delta x^2(t)\rangle/t$. For free-energy profiles with constant drift term, $F(x) \sim ax$, $D_x(t) = \frac{1}{2}\langle (\Delta x - \Delta x_{avg})^2\rangle/t$, where $\Delta x_{avg} = \langle \Delta x\rangle$ is the averages of the corresponding displacements after the time interval $t$. The statistical uncertainties were estimated by bootstrapping.

## Change-point detection in *C. elegans* trajectories

The change points in *Figure 1* (supplemental) were computed using the 'findchangepts' function in MATLAB, which detects the point in a sequence with the maximum difference between the means of values below the point and the mean of the values above the point.

## Acknowledgements

The authors thank Kevin Collins and Sheyum Syed for comments on the manuscript. ADTS is supported by grants from the NSF and NIH. MKa is partially supported by the CHARMM Development Project.

## Additional information

### Funding

| Funder | Grant reference number | Author |
| --- | --- | --- |
| National Science Foundation | BRAIN Initiative EAGER Award | Aravinthan DT Samuel |
| National Institutes of Health | 1P01GM103770 | Aravinthan DT Samuel |
| CHARMM Development Project | | Martin Karplus |

The funders had no role in study design, data collection and interpretation, or the decision to submit the work for publication.

### Author contributions

Mason Klein, Conceptualization, Software, Formal analysis, Investigation, Visualization, Methodology, Writing—original draft, Writing—review and editing; Sergei V Krivov, Conceptualization, Software, Formal analysis, Visualization, Methodology, Writing—original draft, Writing—review and editing; Anggie J Ferrer, Formal analysis, Investigation, Visualization; Linjiao Luo, Formal analysis, Investigation; Aravinthan DT Samuel, Conceptualization, Supervision, Funding acquisition, Writing—original draft, Writing—review and editing; Martin Karplus, Conceptualization, Formal analysis, Supervision, Investigation, Writing—original draft, Writing—review and editing

### Author ORCIDs

Mason Klein ⬢ http://orcid.org/0000-0001-8211-077X

### Decision letter and Author response

Decision letter https://doi.org/10.7554/eLife.30503.014
Author response https://doi.org/10.7554/eLife.30503.015

## Additional files

**Supplementary files**

• Source code 1. Scripts used in conjunction with MAGAT Analyzer software

DOI: https://doi.org/10.7554/eLife.30503.012

• Transparent reporting form

DOI: https://doi.org/10.7554/eLife.30503.013

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
