## [Decision Letter]

Thank you for submitting your article "Exploratory search during directed navigation in *C. elegans* and *Drosophila* larva" for consideration by *eLife*. Your article has been reviewed by two peer reviewers, and the evaluation has been overseen by a Reviewing Editor and K VijayRaghavan as the Senior Editor. The reviewers have opted to remain anonymous.

The reviewers have discussed the reviews with one another and the Reviewing Editor has drafted this decision to help you prepare a revised submission.

Summary:

In this manuscript, the authors attempted to apply the framework of diffusion processes and Markov State Models (MSMs) to analyze the navigational dynamics of *C. elegans* and *Drosophila* larva in an isotropic environment and environments with temperature or salt concentration gradients. They found that in the absence of external stimuli, both organisms follow deterministic dynamics at small time scales, while switch to stochastic (or diffusive) dynamics at long time scales. It was further shown that the introduction of temperature or chemical stimuli has little impact on the diffusive random motion, even though worms eventually navigate towards the favorable environments. Overall, the results provide new insights in understanding navigational dynamics of different organisms and point at possibilities of applying methodologies of analyzing protein folding simulations to these navigational trajectories.

Essential revisions:

The analogy made between the navigation organisms and protein folding should be better explained in the text, especially in the introduction part. The manuscript will be stronger if the authors can elaborate further the link between the search process approaching to non-equilibrium steady state, and the free energy landscape of protein folding, beyond stating that "the navigational dynamics of worms and larvae have some parallels with the complex dynamics of a polypeptide chain navigating to the native structure of the protein to which it corresponds. Both dynamics are stochastic, both need to avoid traps due to local minima, and both were developed by evolution. Hence it is of interest to see whether approaches developed for understanding protein folding dynamics can be used to study the navigational dynamics of worms and larvae.…" Protein folding free energy landscape is rugged and contains numerous metastable states leading to the separation of timescales. For the navigational dynamics of worms, are there also metastable states along the direction of the gradient? Judging from Figure 2' (right panel), it seems that a number of metastable regions do exist. If so, what are the features of these states?

The manuscript needs to further demonstrate the Markovian nature of the studied diffusive and thus the applicability of MSM analysis? When MSMs are applied to protein folding, it is implied that the detailed balance is satisfied due to the reversibility of molecular dynamics simulations. Is this the case also for worms' navigational trajectories? When the authors calculate the transition counts, do they observe a symmetric pattern? In particular, do they observe substantially more counts moving forward than backward along the stimulus gradient (e.g. the first panel in Figure 2)? If their data is largely deviated from the detailed balance, the authors may not obtain faithful estimation of the equilibrium populations.

It would be great if the authors can provide evidence to show that, as stated in the manuscript, there are no active movements. For example, histograms of displacements showing in average there is no net movement will serve this purpose.

The organisms or active particles do reveal biased movements in the short time scale (ballistic motions). The authors should comment on why no biased movements appear in the long time scale (no stimulus), even though there is basically no energetic constraint for the active particles.

*C. elegans* and *Drosophila* larva display different behaviors in their navigational dynamics. For example, the transition from local to global search for *C. elegans* occurs at ~900s (with turn rate reduced by half), while this transition occurs at a longer timescale (relatively constant turn rate). From the biological point of view, could the authors provide some explanation?

---

## [Author Response]

Essential revisions:1) Improve the explanation of the analogy between navigating animals and protein folding.

We have expanded our Introduction section with an additional two paragraphs that make more explicit the connection between protein folding and the navigational dynamics of the two invertebrate systems. We have added more specific examples of traps and free energy landscapes, and noted that this framework has been applied to other systems as well (Introduction, fourth, fifth and sixth paragraphs).

2) Demonstrate in more detail that Markovian description is applicable

We have expanded the Materials and methods section to make the Markovian formulation clearer and shown how to obtain steady state probabilities. In addition (and this is not required for Markovian behavior), we demonstrate that detailed balance is satisfied, which means that equilibrium probabilities are obtained from the analysis. New text in the Materials and methods section explains how this was done by looking at fluxes between bins, and a new figure panel (Figure 2—figure supplement 1) visualizes it (subsection “Determination of steady state and equilibrium probabilities”, second paragraph; Figure 2—figure supplement 1).

3) Evidence of “no active movement” under isotropic conditions4) Comment on why no biased movement appears at long time scales

For all figure panels that show representative trajectories we have added a green arrow that indicates the drift velocity of the population in the x-direction. This denotes a dimensionless drift velocity, normalized to the average speed of animals, and gives the strength/efficiency of navigation. Details of how this is computed (this metric has been used in other work, which we have cited) is included in Materials and methods. We note here that under isotropic conditions the navigation strength is approximately 10x weaker than during thermotaxis or chemotaxis. Individual animals may show biased movement, but the population on average does not (subsection “Diffusion and search patterns under isotropic conditions”, first paragraph; subsection “Video acquisition and behavioral analysis”, last paragraph and Figure 1, Figure 2, Figure 3 and legends).

5) Local → Global search transitions in worms vs. larvae

While we claim that this transition is not as abrupt as previously thought, it is true that it is not observed during the ~1000 s time frame of our larva experiments. We suspect this could be mainly due to much greater mass of larvae compared to worms, and have added this speculation as a paragraph in the Discussion section, along with an idea of how this could be measured in *Drosophila* with systematically starved animals (Discussion, fourth paragraph).